# Consumption of Sugary Drinks among Urban Adults in Colombia: Association with Sociodemographic Factors and Body Adiposity

**DOI:** 10.3390/ijerph20043057

**Published:** 2023-02-09

**Authors:** Maria A. Santana-Jiménez, Luz D. Nieves-Barreto, Angélica Montaño-Rodríguez, Carolina Betancourt-Villamizar, Carlos O. Mendivil

**Affiliations:** 1School of Medicine, Los Andes University, Bogotá 110111, Colombia; 2Team Foods, Carrera 11 # 84-09, Bogotá 110001, Colombia; 3Section of Endocrinology, Fundación Santa Fe de Bogotá, Bogotá 111071, Colombia

**Keywords:** sugar-sweetened beverages, chronic diseases, obesity, diabetes, soda, nutrition

## Abstract

Introduction: Calories from sugar-sweetened beverages (SSBs) contribute to the development of noncommunicable diseases. There is limited knowledge of the intake of SSBs and their correlates in developing countries. Thus, this study aimed to estimate the consumption of multiple SSBs and their sociodemographic correlates in an urban adult population from Colombia, South America. Methods: This was a probabilistic, population-level study of adults aged 18 to 75 from five cities representing different regions of Colombia. Dietary intake was assessed employing a 157-item semiquantitative food frequency questionnaire that inquired about intake over the last year. The consumption of regular soda, low-calorie soda, homemade and industrialized fruit juices, energy drinks, sport drinks, malt drinks and traditional sugar cane infusion (“*agua de panela*”) was analyzed for the total sample and subgroups defined by sociodemographic and clinical factors of interest. Results: The study included 1491 individuals (female: 54.2%, mean age: 45.3, overweight: 38.0%, obese: 23.3%). Sugary beverages contributed, on average, 287 Cal/d among women and 334 Cal/d among men, representing 8.9% of total daily calories (TDC). Women in the lowest SEL consumed 10.6% of their TDC from sugary drinks, as opposed to 6.6% for those in a high SEL. For men, this difference was not present (*p-value* for interaction = 0.039). Interestingly, a higher educational level correlated with a lower consumption of calories from sugary drinks only among men. Fruit juices were by far the main source of sugary drinks, and their consumption did not change sizably by sex and socioeconomic or educational level. Among women, there was a negative association between socioeconomic level (SEL) and consumption of regular soda, a 50% difference between extreme levels. The intake of low-calorie soda was much higher among men than women, and it more than tripled in the highest vs. lowest SEL among men. The consumption of energy drinks was heavily concentrated in men of low SEL. Conclusion: Colombian urban adults obtain a considerable proportion of their calories from sugary drinks, especially vulnerable groups such as women with lower education. Given the recent acceleration of the obesity epidemic in Latin America, strategies to limit the intake of such liquid calories may provide important public health benefits.

## 1. Introduction

Over the last few decades, rapid changes in lifestyle have led to an increase in the societal burden of diseases such as overweight and obesity, especially in developing countries [1]. By 2016, an estimated 39% of the world’s adult population was overweight and 13% was obese. This represents a three-fold increase since 1975 [2]. As a result, both morbidity and mortality from chronic diseases have risen steadily, constituting a serious problem in countries where economic resources are scarce, and hunger and malnutrition are still relevant concerns [3]. Importantly, the prevalence of overweight and obesity is higher than that of undernutrition in most developing countries [4]. In Latin America, the prevalence of obesity ranges between 24.4% and 43.2% in different countries [5,6]. There is great concern not just about the size of the obesity problem but also the expected long-term increase in chronic diseases that will certainly ensue. Part of the societal effort to mitigate the burden derived from obesity involves a detailed study of the multiple dietary factors that may influence body weight across the life course and the analysis of their distribution among population subgroups.

Recent changes in lifestyle in Latin American countries have included a substantial increase in the intake of added sugars, particularly from beverages [1]. Evidence from multiple studies has documented a higher incidence and prevalence of excess body weight among regular consumers of sugar-sweetened caloric beverages (SSBs), compared to nonconsumers [7,8]. Despite the contradictory evidence on whether added sugars are more obesogenic than other nutrients under isocaloric comparisons, several mechanisms pose added sugars as a desirable target for obesity prevention. SSBs, and more broadly all foods rich in refined carbohydrates or added sugars, are able to set in motion reward mechanisms in the central nervous system (CNS) analogous to those induced by the consumption of psychoactive drugs. Due to the fact of their fast absorption and prompt delivery of simple carbohydrates to the CNS, SSBs are particularly prone to stimulate reward signals and contribute to “food addiction” [9]. In fact, sugar rewards can be comparable to, and sometimes even stronger than, those present in individuals with problems of substance dependence or abuse.

In addition to excess body weight, SSBs are linked to a heightened risk of cardiovascular diseases [10]. Several mechanisms explain the relationship between SSBs consumption, weight gain, and chronic diseases: poor satiety, excessive caloric content, and incomplete nutritional compensation at subsequent meals among others [11].

Even though the increase in the sugar content of modern diets is widely known, evidence addressing the consumption of sugary beverages in developing nations is scarce. Data from advanced economies such as the United States show a steady decline in SSB consumption over the recent years [12], but such knowledge is currently unavailable for most Latin American countries. In order to guide public health policy, it is critical to understand how SSB consumption behaves in low- and middle-income countries (LMICs) and the population segments in which it is concentrated. This is time-sensitive information, as these beverages continuously gain in popularity due to their lower prices and aggressive marketing [13].

Some of the most important correlates of food and nutrient consumption are sex, age, socioeconomic level (SEL), and education [3]. Several studies have found associations between SSB consumption and younger age, lower education, a high density of establishments selling SSBs, and lack of the availability of fruits and vegetables in nearby supermarkets [14]. Nevertheless, most of these studies analyzed aggregated SSB data, and a more detailed understanding of the factors linked with each specific SSB intake in each country or community is needed to inform the design of tailored, successful public policies.

Within this context, this study aimed to evaluate the association of SSBs with demographic variables, SEL, education, and body mass index (BMI) in a population-based sample of adults from Colombia, South America. A secondary aim was to suggest interventions based on the study results that may increase the success of public policies aimed at moderating SSB intake among adults from Colombia and other countries of similar characteristics.

## 2. Materials and Methods

The study took place in Colombia, a Latin American country with an estimated population of 48 million, 77% of whom live in urban areas [15]. Approximately 30% of the Colombian population resides in the five cities included (Bogotá, Medellín, Barranquilla, Cali and Bucaramanga).

### 2.1. Sampling and Data Collection

This analysis was based on data from COPEN (*Estudio Colombiano de Perfiles Nutricionales*—Colombian Study of Nutritional Profiles), a population-based, cross-sectional, multistage sampling study that included one large city from each of Colombia’s five major regions. The sampling frame was obtained from the 2005 census of the Colombian population [16], cartography was obtained from the national geostatistical frame developed by the Colombian National Department of Statistics and data on socioeconomic stratum (SES) came from the National Superintendence of Public Services. The first stage of sampling selected the cartographic sectors, within sectors blocks were selected (on average 8 per cartographic sector), within blocks households were selected, and within households individual participants were selected. The sample was stratified by city, sex, age group and socioeconomic stratum of the household.

The study was executed between June and November of 2018. Information was captured using a tablet device containing digital forms with proper validation rules, developed for the study. All staff in charge of data collection was extensively trained by the study’s Principal Investigator. With this design and including the design effect, the study sample yielded an overall sampling error of 2.2% for the prevalence of overweight or obesity in the target population, which was a central objective of the COPEN study. The sampling errors for each city were, respectively, Bogotá 4.0%, Medellín 5.0%, Cali 5.0%, Barranquilla 5.6% and Bucaramanga 6.8%.

### 2.2. Participants

For the effects of this report, the participants were individuals between the ages of 18 and 75 and living in one of the five cities mentioned above. Foreigners living in Colombia, individuals in hemodialysis or peritoneal dialysis therapy and persons with disabilities that precluded a reliable fulfillment of the study questionnaire were excluded from participation.

### 2.3. Sociodemographic and Anthropometric Variables

Information on sex, date of birth, household SES, marital status, individual educational level and employment status was collected using a standardized questionnaire. SES is classified in Colombia by the National Statistics Department in 6 strata according to the characteristics of the residence (with stratum 1 being the lowest and stratum 6 being the highest). Residential dwellings are classified according to their physical characteristics and environment. The methodology for this classification creates homogeneous strata taking as input information concerning land use, public utilities, access routes, topography, land valuation and property characteristics. This information is very well established and freely accessible [17]. It also has a significant correlation with household income. Given that sociodemographic, income and human development indicators are more similar for individuals living in strata 4 to 6 than among the other strata [17], SES was analyzed in three groups, corresponding to strata 1–2 (low SES), 3 (medium SES) and 4–6 (high SES). The participants were asked to report the highest educational cycle they had completed: preschool, primary school, secondary school, technical degree, college degree or post-graduate degree. For the statistical analyses, the variable educational level was operationalized into three categories: elementary or lower, secondary or technical degree and college or higher. Height was measured using a portable stadiometer supported on a firm surface. Weight was measured employing a solar digital scale with 100 g sensitivity and 200 Kg capacity.

### 2.4. Food Frequency Questionnaire

Dietary intake was assessed using a semiquantitative food frequency questionnaire (FFQ) with a 157-item food list plus the frequency of intake and number of standard portions consumed (with the reference portion size written next to this field). This FFQ had been previously developed and piloted in the Colombian population [18]. The portion sizes were calculated according to the coding of weights and measurements in the National Survey of Nutritional 2005 (ENSIN 2005), and the unit of measure most frequently reported for each food was used as the reference portion size. The average frequency of intake for each food item over the last year was registered as one of nine categories: never, 1–3 times per month, once a week, 2–4 times a week, 5–6 times a week, once a day, 2–3 times a day, 4–6 times a day or more than 6 times a day. A trained staff member administered the FFQ and registered all the information.

Food composition data were obtained from the Colombian Institute of Family Welfare (ICBF—Instituto Colombiano de Bienestar Familiar) reference tables [19] or the US Department of Agriculture food composition database (USDA, FoodData Central). For foods not represented in any of these sources, information from the manufacturer was employed. The foods included in the analyses reported in this paper were regular soda, low-calorie (diet) soda, homemade fruit juices, industrialized fruit juices, energy drinks, sport drinks, malt drinks and traditional sugar cane infusion (“*agua de panela*”).

### 2.5. Data Analysis

All estimations were projected to the target study population using city, sex, age group and SES-specific expansion factors according to the study’s multistage sampling design. The mean daily calories from SSBs and the mean daily servings of each SSB were compared across categories of ordinal predictors using a one-way linear model (ANOVA). When the global ANOVA was significant, post hoc pairwise comparisons were performed against a reference category (the lowest) using Dunnett’s method. All analyses were 2-tailed and carried out at a 5% significance level. All analyses were performed in SPSS for Windows, v.21 (Cary, NC, USA).

### 2.6. Ethical Aspects

The study was conducted in accordance with the Declaration of Helsinki and with national and international regulations. All participants provided written informed consent. The study was approved by the IRB of Universidad de los Andes (*Comité de Ética de la Vicerrectoría de Investigaciones*) according to minute 1016 of 27 April 2018.

## 3. Results

The study sample comprised 1491 adults (54% women) with a mean age of 45.3 years and mean BMI of 25.8 Kg/m2, 61.3% of them had overweight or obesity. Most participants were in the low SEL category and had high school or technical education (Table 1).

The mean daily consumption of calories from sugary drinks was significantly higher among men than women (334 vs. 287 Cal/d on average, *p* = 0.017) and decreased steadily across age groups in both sexes (*p* < 0.001) (Figure 1). Consumption was also considerably lower in the high SEL compared to low (*p* = 0.002), especially for women. The intake of calories from sugary drinks decreased with higher education (*p* = 0.034) but in this case more pronouncedly in the male sex. Both men and women with excess body weight reported less calories from sugary drinks (*p* = 0.007 for overweight vs. low/normal, *p* = 0.024 for obesity vs. low/normal).

Regular soda was similarly consumed in both sexes and was not significantly associated with SEL, educational level or body mass index category (*p* > 0.05 for all). There was a significantly lower intake of regular soda with increasing age group in both sexes (*p* < 0.001 in both cases) (Figure 2). Despite a trend towards a higher intake of regular soda in women with a lower household SEL or higher education, none of these differences reached statistical significance.

Compared to regular soda, the mean consumption of low-calorie soda (LCS) was much lower in all subgroups. Men consumed, on average, three times more LCS than women (*p* = 0.003), a difference most pronounced in the 40–59 age group. As opposed to regular soda, LCS intake increased monotonically with SEL (*p* = 0.051) but only modestly and nonsignificantly with educational level (Figure 3). Despite a substantially higher consumption of LCS by men with overweight or obesity, there was no significant difference by BMI category (*p* = 0.10).

The consumption of homemade fruit juice was similarly high across age, SEL, education and BMI-defined categories in both sexes (Figure 4), with no statistically significant differences.

Lastly, the consumption of traditional sugarcane infusion (*agua de panela*) was significantly lower in the high compared to low SEL (39% lower, *p* = 0.01), with a more pronounced difference among women (46.2% between extreme SEL categories). A similar trend was observed across educational levels, with a 59% lower intake among college graduates relative to those with only primary schooling (*p* < 0.001). The consumption of *agua de panela* did not vary by sex.

## 4. Discussion

This population-based study of urban adults from Colombia analyzed the distribution and correlates of multiple sugar-sweetened drinks. SSBs were an important source of energy, representing 8.9% of total daily calories in both sexes. Fruit juices were by far the main source of sugary drinks in all age groups and population segments. Conversely, the consumption of other SSBs changed notoriously according to sex, education, and SEL.

Despite the existence of prior literature on the consumption of SSBs in developing countries, there was very limited evidence coming from Latin America, particularly from Andean countries. There was also an insufficient understanding of how factors that are often beyond personal choice, such as sex, education, and socioeconomic position, may be determinants or correlates of SSB consumption and, hence, impact dietary quality and the risk for noncommunicable diseases in this context. In addition, it is essential to examine the contribution of each specific SSB to total intake, and to what extent their consumption is concentrated in particular population segments.

The absolute intake of calories from SSBs was higher in men compared to women. Nonetheless, the difference disappeared when adjusted for total calories, so it was most likely due to the fact of larger body sizes and higher energy requirements among males. Many potential factors explain why the consumption of SSBs decreased for both sexes across age groups. First, the use of electronic devices (“screen time”) tends to be higher among younger people, and screen time is positively and significantly associated with daily SSB consumption [20]. The underlying explanation may involve the direct marketing of SSBs to young adults, which is highly prevalent and is now shifting from TV toward social networks accessible through smartphones, tablets, and computers [21]. Second, in recent times there is increasing concern regarding healthy aging issues among urban adults, which may promote positive behavioral changes [22]. Lastly, there are economic considerations at play in this phenomenon. In Colombia and other Latin American countries, many people aged 18 to 39 are still in the process of reaching economic independence [23]. Therefore, SSBs are attractive due to the fact of their low prices and ubiquitous availability [13].

Interestingly, the consumption of SSBs decreased with greater SEL only for women and most notoriously in the highest SEL. Almost all of this difference occurred at the expense of regular soda, the intake of which was approximately half the amount in this SEL compared to the lowest. Women of high SEL have greater monetary resources to purchase other beverages, but also the density of neighborhood stores (“*tiendas de barrio*”) selling SSBs tends to be much lower in high-income areas. Both the density of stores and the distance to one of them are important correlates of SSB consumption [14]. In women, this pattern was not observed when educational level was the independent variable.

On the other hand, educational level *was* an important predictor of the SSB intake of men. Higher education correlated with lower SSB consumption so that only men of high education approached the lower level observed in women. Several factors may be at play behind this pattern. First, men with low education may have less knowledge concerning the health impacts of SSBs. Second, they usually have physically demanding jobs with long hours and a high stress environment [24], all of which favor the consumption of SSBs, particularly energy drinks. Our results on energy drinks support this hypothesis. The reason why this trend was not evident among women may be linked to cultural factors that make females in Latin America generally more concerned with body image and self-care [25]. An unexpected finding was that individuals with excess body weight reported a lower intake of calories from SSBs. This may reflect intentional caloric restriction as an attempt to lose weight, but it may also be a consequence of inaccurate reporting of SSB intake among overweight individuals. This issue will have to be explored in greater detail in future studies. The cross-sectional nature of the study precludes us from reaching conclusions concerning the contribution of SSBs to the genesis of excess body weight in our population, but future waves of COPEN will allow us to examine this hypothesis in greater detail.

Our results showed that LCS were consumed much less than regular sodas in all population subgroups, perhaps because they are newer products still growing in the local market. The intake of LCS was also considerably higher among men relative to women. To explain this, it is important to mention that LCS are artificially sweetened beverages, and some population subgroups hold the strong belief that natural sweeteners are inherently healthier than artificial ones [26]. One such group is that of urban females [27]. As opposed to what occurred with regular soda, the consumption of LCS showed a positive correlation with SEL. Cost did not seem to be the main reason, as LCS are equally priced or even slightly cheaper than regular sodas in Colombia. Therefore, the availability, cultural factors and perceptions concerning the health properties of beverages may underlie this pattern. It is essential to dispel common misconceptions regarding SSBs, such as the widely held belief that SSBs with a “natural” origin, such as fruit juices, are inherently healthy or that any beverage that does not contribute any sugar or calories but is of “artificial” origin is necessarily unhealthy. The focus should be placed on curtailing the overall content of added and liquid sugars in the diet, regardless of their origin.

In contrast to our findings, other researchers have found a greater intake of low-calorie sodas among women in the United States [28]. Nonetheless, the cause for this discordance may be the interplay between culture, SEL and sex in the preference for regular versus low-calorie sodas [29]. In our study population, the sex preferences seem to differ from what was previously reported.

Some SSBs are closely related to Colombian cultural traditions, such as homemade fruit juice and sugarcane infusion *(agua de panela).* For homemade fruit juices, in particular, all population subgroups had a similarly high intake, meaning that regardless of sociodemographic characteristics, fruit juice is still the most popular drink around the country. This is also consistent with the conception that “natural is always healthier”, a misconception that leads people to not realize that homemade fruit juices are a considerable source of sugar due the fact of their fructose content and the loss of the fiber present in the whole fruit [30,31]. As expected, the consumption of *agua de panela* was by far highest at the lowest socioeconomic and educational levels. This is an inexpensive, easy-to-make, and culturally popular beverage, but it provides approximately 20 g of sugar—as sucrose—and 80 calories per small serving [18].

### 4.1. Implications for Intervention

Our results show that in the local context, successful interventions aimed at curbing SSB intake may induce close to a 10% reduction in caloric intake at the population level. Such a reduction would translate into sizable changes in the caloric balance, the population distribution of BMI and likely into reductions in the risk of multiple adverse health outcomes in the long term. The results also suggest that for maximum impact, such interventions should be targeted at women of low SEL and at men of a relatively lower educational level. One key result was that fruit juices and *agua de panela* were extremely relevant sources of liquid calories, and most educational campaigns on SSBs have focused entirely on industrial SSBs, particularly soft drinks. In order to substantially impact total SSB consumption, educational and regulation efforts need to be broadened to include other SSBs traditionally considered by the general population as healthy. Further, the low penetrance and virtually null caloric content of LCS suggests that they could be useful as part of a wider dietary portfolio of options to replace SSBs.

### 4.2. Strengths and Limitations

COPEN was designed on a probabilistic, population-based sample of the main Colombian cities and interrogated usual intake over the last year, as opposed to just recent intake. The study inquired individually about regular and low-calorie sodas, homemade fruit juices, sport drinks, energy drinks and traditional sugary beverages, allowing it to reach more detailed conclusions concerning the relevant correlates of each type of SSBs, which turned out to be considerably different. The data collection was carefully performed by well-trained personnel using standardized procedures, and several quality checks were performed on the raw data. One of the central shortcomings of the study is the absence of rural representation, which naturally limits its overall generalizability. Logistic and cost considerations weighed in the decision of concentrating on urban centers, but we do wish to extend future versions of the study to rural and semirural areas, where the situation concerning SSBs may be quite different from what we found in cities. In addition, an important limitation is the absence of information on comorbidities that may be associated with SSB intake. Being a population-based study focused on dietary and lifestyle factors, COPEN did not collect extensive medical information from the participants. Yet another relevant limitation is that the data were not entirely current, and the COVID-19 pandemic may have influenced dietary intake in ways that could not be predicted at the time of the study execution. Nonetheless, we plan on undertaking new waves of COPEN, and a detailed examination of the time trends in SSBs consumption will definitely be one of the main topics of subsequent analyses. The examination of such trends will inform about successful and unsuccessful strategies to hold back SSB consumption in our countries.

## 5. Conclusions

In summary, our results show that SSBs contribute considerably to the total energy balance of Colombian adults, and their consumption is strongly determined by basic sociodemographic factors. The exception was homemade fruit juice, the intake of which was equally high in essentially all population subgroups. A better social position, manifested by a better SEL or a higher educational level, was associated with the reduced intake of SSBs only in women. This information should inform the design of policies to limit the intake of SSBs, including the tailoring of campaigns aimed at a masculine public, the introduction of well-designed taxes that preferentially impact SSBs and mass education on the health effects of fruit juices, whether or not they have added sugars.

## Figures and Tables

**Figure 1 ijerph-20-03057-f001:**
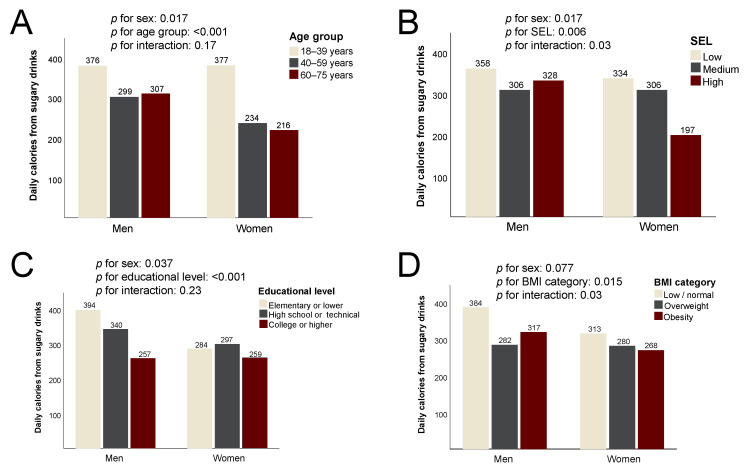
Total daily consumption of calories from sugary drinks among the study participants. Data are mean daily Calories from sugar-sweetened beverages according to (**A**) age group; (**B**) socioeconomic level (SEL); (**C**) educational level; (**D**) body mass index (BMI) category.

**Figure 2 ijerph-20-03057-f002:**
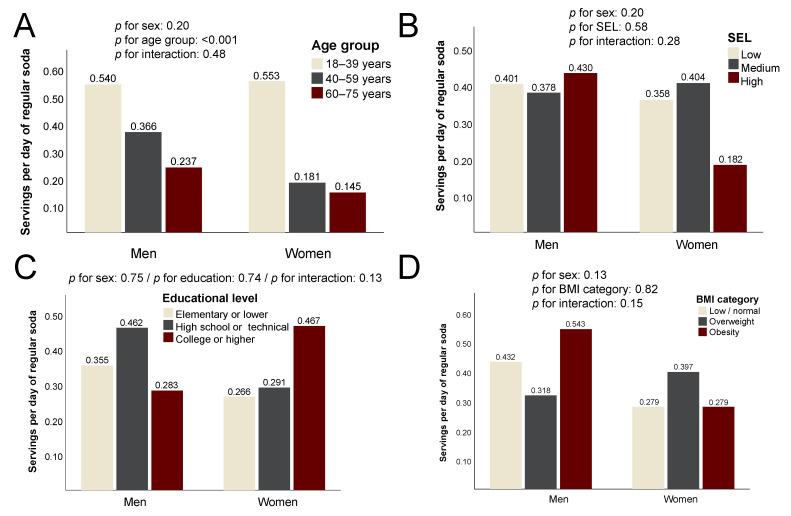
Daily consumption of regular soda among the study participants. Data are the mean servings per day according to (**A**) age group; (**B**) socioeconomic level (SEL); (**C**) educational level; (**D**) body mass index (BMI) category.

**Figure 3 ijerph-20-03057-f003:**
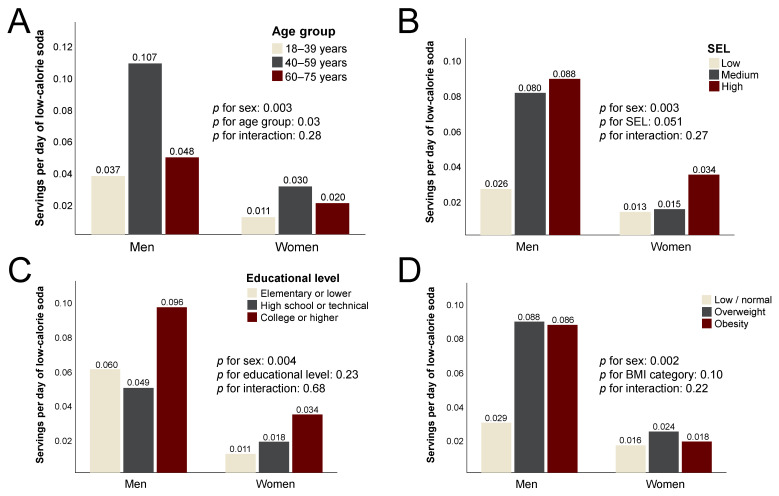
Daily consumption of low-calorie soda among the study participants. The data are the mean servings per day according to (**A**) age group; (**B**) socioeconomic level (SEL); (**C**) educational level; (**D**) body mass index (BMI) category.

**Figure 4 ijerph-20-03057-f004:**
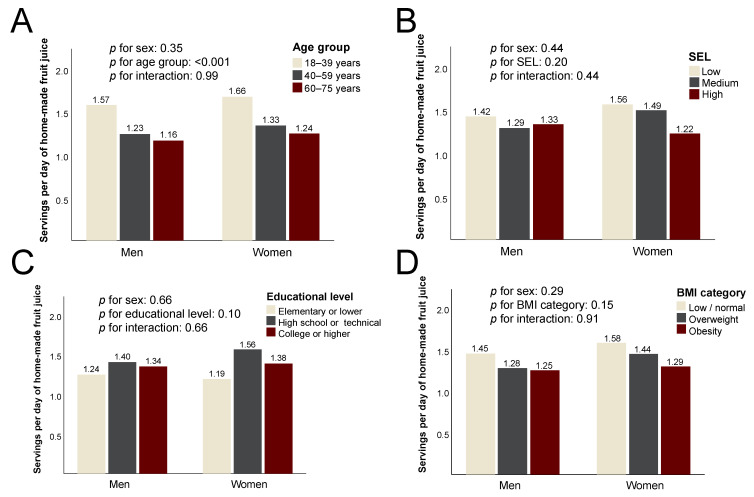
Daily consumption of homemade fruit juice among the study participants. The data are the mean servings per day according to (**A**) age group; (**B**) socioeconomic level (SEL); (**C**) educational level; (**D**) body mass index (BMI) category.

**Table 1 ijerph-20-03057-t001:** Characteristics of the study participants.

	Men (*n* = 682)	Women (*n* = 809)	Total (*n* = 1491)
Age (yrs)	44.7 ± 18.27	45.8 ± 17.06	45.3 ± 17.63
**Age group**			
18–39	286 (41.9%)	325 (40.2%)	611 (41.0%)
40–59	204 (29.9%)	264 (32.6%)	468 (31.4%)
60–75	192 (28.2%)	220 (27.2%)	412 (27.6%)
Weight	74.5 ± 15.4	67.2 ± 13.3	70.6 ± 14.8
Height	169.7 ± 7.2	155.7 ± 8.0	162.1 ± 10.3
Body mass index	25.8 ± 4.9	27.7 ± 5.4	25.8 ± 5.3
% Overweight	39.7	36.5	38.0
% Obesity	14.5	30.7	23.3
**Socioeconomic level**			
Low	271 (39.7%)	347 (42.9%)	618 (41.4%)
Medium	207 (30.4%)	230 (28.4%)	437 (29.3%)
High	204 (29.9%)	232 (28.7%)	436 (29.2%)
**City**			
Barranquilla	109 (16.0%)	122 (15.1%)	231 (15.5%)
Bogotá	219 (32.1%)	261 (32.3%)	480 (32.2%)
Bucaramanga	70 (10.3%)	94 (11.6%)	164 (11.0%)
Cali	137 (20.1%)	168 (20.8%)	305 (20.5%)
Medellín	147 (21.6%)	164 (20.3%)	311 (20.9%)
**Educational level**			
Elementary or lower	137 (20.1%)	177 (21.9%)	314 (21.1%)
High school or technical	402 (58.9%)	472 (58.3%)	874 (58.6%)
College or higher	143 (21.0%)	160 (19.8%)	303 (20.3%)

## Data Availability

The data presented in this study are available upon request from the corresponding author. The data are not publicly available due to the privacy of the included subjects.

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
