# Peer review of "Consumption of Sugary Drinks among Urban Adults in Colombia: Association with Sociodemographic Factors and Body Adiposity"

_ijerph, 2023, doi:10.3390/ijerph20043057_

Round 1

Reviewer 1 Report

The study of Santana Jiménez et al seeks for an association between the level of sugar consumption and some socio-demographic characters of a sample in Colombia. The number of participants and design of the study are appropriate.

I have some concern about the novelty of this study. Different studies have been carried out on the same topic, reaching similar conclusions. The authors are encouraged to highlight the novelty of their study in the context of existing literature, supporting the need of this research.

Some minor considerations:

Please, state the aim of the study in the abstract.

For the sake of clarity, the authors are invited to evidence the significance of data in the graphs provided.

The paper should undergo a minor language editing.

Reviewer 2 Report

A very interesting paper about a real problem during the XXIst century, the role of sugary drinks in several health problems has been significant last 20 years. I would like to make some comments trying to improve the quality of the paper:

- I would like to make a global suggestion to write the paper in Third person, for sure I know it's possible to make it in first person but I highly recommend to use the Third one.

- Abstract is too long (more than 350 words), it can be improved, the description of the background and especially about the method are not really clear (about the method/how the authors have done the study, there is no real explanation).

- The study was executed between June and November of 2018, so the data are not really actual data. Please, explain why it wasn't presented previously and possible differences and biases with actual situation of SSB consumption.

- There are no secondary/operative aims about how to transfer the results to practice (a common limitation in research papers).

- Methodology is well descripted, congratulations to the authors.

- The sentence "Unexpectedly, men and women with excess body
weight reported less Calories from sugary drinks (p=0.007 for overweight vs low/normal,p=0.024 for obesity vs low/normal)", includes an opinion of the authors (Unexpectedly..:). Please, use this sentence in discusion part more than Results.

- A big limitation in the study is the lack of comorbidity analysis about disorders that can interfer in the use of SSB or can be consequent of them, there is no mention about the influence of several problems about the topic and the prevalence on consumption.

There is no limitations part in the paper, and I think it can be fundamental to clarify several aspects of the internal and external validity of the research (sample, method, tools...).

- Also there are no practical implications derivated from the research, about a very actual and interesting topic. Please, try to purpose something new and connected with the data.

Round 2

Reviewer 1 Report

I appreciated the Authors' efforts in reviewing the manuscript, substantially. The overall quality is definitively improved although the introduction would benefit from citing key articles to properly frame the theoretical background, such as:

10.1016/j.jad.2016.10.035

Author Response

Thank you for assessing the revised manuscript. We have included a paragraph in allusion to the suggested paper and cited it (page 3, second paragraph).